# Nonsteroidal Mineralocorticoid Receptor Antagonism by Finerenone—Translational Aspects and Clinical Perspectives across Multiple Organ Systems

**DOI:** 10.3390/ijms23169243

**Published:** 2022-08-17

**Authors:** Peter Kolkhof, Robert Lawatscheck, Gerasimos Filippatos, George L. Bakris

**Affiliations:** 1Cardiology Precision Medicines, Research & Early Development, Bayer AG, Aprather Weg 18a, 42113 Wuppertal, Germany; 2Clinical Development, Bayer AG, Müller Straße 178, Building P300, 13342 Berlin, Germany; 3Department of Cardiology, School of Medicine, National and Kapodistrian University of Athens, Attikon University Hospital, Mikras Asias 75, 115 27 Athina, Greece; 4Department of Medicine, University of Chicago Medicine, 5841 S. Maryland Ave., Chicago, IL 60637, USA

**Keywords:** aldosterone, cardiorenal, cardiovascular, fibrosis, finerenone, hypertrophy, inflammation, kidney, mineralocorticoid receptor, oxidative stress

## Abstract

Perception of the role of the aldosterone/mineralocorticoid receptor (MR) ensemble has been extended from a previously renal epithelial-centered focus on sodium and volume homeostasis to an understanding of their role as systemic modulators of reactive oxygen species, inflammation, and fibrosis. Steroidal MR antagonists (MRAs) are included in treatment paradigms for resistant hypertension and heart failure with reduced ejection fraction, while more recently, the nonsteroidal MRA finerenone was shown to reduce renal and cardiovascular outcomes in two large phase III trials (FIDELIO-DKD and FIGARO-DKD) in patients with chronic kidney disease and type 2 diabetes, respectively. Here, we provide an overview of the pathophysiologic role of MR overactivation and preclinical evidence with the nonsteroidal MRA finerenone in a range of different disease models with respect to major components of the aggregate mode of action, including interfering with reactive oxygen species generation, inflammation, fibrosis, and hypertrophy. We describe a time-dependent effect of these mechanistic components and the potential modification of major clinical parameters, as well as the impact on clinical renal and cardiovascular outcomes as observed in FIDELIO-DKD and FIGARO-DKD. Finally, we provide an outlook on potential future clinical indications and ongoing clinical studies with finerenone, including a combination study with a sodium–glucose cotransporter-2 inhibitor.

## 1. Introduction

The mineralocorticoid receptor (MR) is activated by aldosterone and cortisol, and it belongs to the steroid hormone receptor subfamily of nuclear hormone receptors that function as ligand-activated transcription factors to directly affect gene expression [1,2]. Aldosterone selectivity for the MR is conferred by the 11 beta hydroxysteroid dehydrogenase enzyme 2 (11-β HSD2), which converts cortisol into receptor-inactive cortisone, enabling the preferred aldosterone occupancy of MR and subsequent activation [2,3,4]. The MR is expressed across a range of organ systems in the body and has been documented at the organ level in the kidney, heart, vasculature, colon, brain, eye, skin, lung, liver, skeletal muscle, adipose tissue, salivary glands, and sweat glands [2,5,6]. In the kidneys, MR expression has been observed in epithelial, tubular, and glomerular cells, including podocytes and mesangial cells [6,7]. In the cardiovascular system, the MR has been detected in cardiomyocytes and cardiac fibroblasts, as well as in endothelial cells (ECs) and vascular smooth muscle cells (SMCs) [1,8]. The MR can also be found in other cells, including myeloid cells, immune cells, adipocytes, and fibroblasts [2,5,9].

In the kidneys, MR primarily acts as a regulator of fluid and electrolyte homeostasis in conjunction with aldosterone [1]. However, the MR also plays a role in normal physiology, vascular function, and tissue repair and remodeling [10]. The inappropriate overactivation of the MR is caused by increased aldosterone and the elevated expression of the MR. Overactivation of the MR can occur independently of aldosterone through activation by cortisol in addition to ligand-independent activation via the small GTPase Ras-related C3 botulinum toxin substrate 1 (Rac1) [11,12]. When the MR is overactivated, the expression of pro-inflammatory and pro-fibrotic genes are upregulated in their respective cells and organs, resulting in organ dysfunction and damage. Therefore, MR overactivation is implicated in the pathophysiology of renal and cardiovascular diseases [13]. In the kidneys, MR overactivation can result in fibrosis, glomerulosclerosis, a decline in glomerular filtration rate (GFR), proteinuria, and tubulointerstitial damage [1,14], whereas MR overactivation in the heart can cause myocardial hypertrophy, fibrosis, inflammation, and oxidative stress [15]. Moreover, it has been demonstrated that aldosterone causes the rapid expression of cardiac voltage-operated Ca^2+^ T-channels and thereby has a chronotropic effect in cultured neonatal rat ventricular myocytes [16]. This rapid electrical remodeling could contribute to the deleterious effect of hyperaldosteronism on cardiac function [17]. In addition, vascular damage as a result of MR overactivation can manifest as inappropriate vascular remodeling, endothelial dysfunction, and increased vascular stiffness [15].

MR overactivation can be inhibited by two different classes of MR antagonists (MRAs). Steroidal MRAs, including spironolactone and eplerenone, are well-established therapies for heart failure (HF) with reduced ejection fraction (HFrEF) and resistant hypertension due to primary aldosteronism [13,15,18]. However, the use of steroidal MRAs is limited because of the increased risk of hyperkalemia, as well as unwanted sexual side effects such as breast pain and gynecomastia [19]. The second novel class of MRAs are the nonsteroidal MRAs, which were designed to target the MR while maintaining an acceptable risk–benefit profile [1,5]. Finerenone is a novel, selective, nonsteroidal MRA that is efficacious in patients with type 2 diabetes (T2D) and chronic kidney disease (CKD) [20,21]. The phase III clinical trial program for finerenone included two pivotal trials that investigated the renal and cardiovascular benefits of finerenone across a broad spectrum of CKD and T2D. Together, these trials form the largest cardiorenal outcome program in this patient population to date [22]. In the FInerenone in reducing kiDnEy faiLure and dIsease prOgression in Diabetic Kidney Disease (FIDELIO-DKD) study, finerenone significantly reduced the risk of the renal composite outcome and the key secondary cardiovascular composite outcome in patients with predominantly stage 3–4 CKD and severely increased albuminuria and T2D [20]. In the FInerenone in reducinG cArdiovascular moRtality and mOrbidity in Diabetic Kidney Disease (FIGARO-DKD) study, finerenone significantly reduced the primary cardiovascular composite outcome risk in a less advanced CKD in T2D patient population than that studied in FIDELIO-DKD (patients with stage 2–4 CKD and moderately increased albuminuria or stage 1–2 CKD with severely increased albuminuria) [21]. The FInerenone in chronic kiDney diseasE and type 2 diabetes: Combined FIDELIO-DKD and FIGARO-DKD Trial programme analYsis (FIDELITY), a prespecified pooled analysis of the two trials, has since provided robust evidence for the cardiorenal benefits across the spectrum of CKD severity in >13,000 patients [22].

Preclinical evidence describing the mode of action of steroidal and nonsteroidal MRAs in models of cardiorenal disease has been previously reviewed [23]; however, evidence on the components of the mechanism of action of finerenone across a variety of organ systems is rapidly growing. The objective of this review is to provide a comprehensive overview of current knowledge of the pathophysiologic role of MR overactivation and preclinical evidence with the nonsteroidal MRA finerenone in a range of disease models and organ systems. In addition, mechanistic explanations for the cardiorenal benefits observed with finerenone in FIDELIO-DKD and FIGARO-DKD are postulated, and potential future clinical indications for finerenone are also explored.

## 2. The Effects of Finerenone’s Mechanism of Action 

The available evidence suggests that finerenone offers cardiorenal protection, as observed in patients with HF and mild-to-moderate CKD (phase II study MinerAlocorticoid Receptor antagonist Tolerability Study (ARTS)) and T2D and CKD (phase III studies FIDELIO/FIGARO; phase II study MinerAlocorticoid Receptor antagonist Tolerability Study in Diabetic Nephropathy (ARTS-DN)) via a combination of different mechanisms determined in preclinical studies. As an MRA, finerenone acts as a natriuretic [1] to prevent sodium and fluid retention in the body and, thus, the development of hypertension [24]. By blocking the MR, finerenone may also inhibit the generation of reactive oxygen species (ROS), which promote oxidative stress in cells of the kidney [25,26], cardiac, and vascular systems, leading to tissue injury. Finerenone also appears to prevent inflammation and fibrosis driven by the MR on inflammatory cells, which further contribute to tissue damage, as well as hypertrophy and tissue remodeling, in the cardiovascular system [19,27,28].

### 2.1. Sodium Retention

Under normal physiologic conditions in the distal nephron of the kidney, aldosterone promotes sodium reabsorption and water retention, as well as potassium and magnesium excretion via the MR to regulate extracellular volume and blood pressure [24]. MR activation in renal epithelial cells leads to the gene expression of subunits of epithelial sodium channels, the upregulation of serum- and glucocorticoid-regulated kinase 1 (Sgk1), and finally, increased sodium transport in epithelial tissues. In the cell membrane, the turnover of the sodium channels is mediated by the neural precursor cell-expressed developmentally down-regulated 4 ligase (Nedd4-2), a ubiquitin protein ligase. The phosphorylation of Nedd4-2 by Sgk1 prevents the binding of Nedd4-2 to channels, thereby promoting sodium influx [29].

Natriuresis is beneficial in patients with cardiovascular diseases because it can lower blood pressure and reduce the risk of myocardial infarction (MI) and stroke. MRAs are able to inhibit sodium retention, and finerenone has demonstrated dose-dependent natriuretic efficacy in healthy human volunteers [1,30]. Neurohumoral stimulation by the renin–angiotensin–aldosterone system, the sympathetic nervous system, and vasopressin contribute to permanent sodium retention and an associated extracellular volume load in the development of chronic HF and kidney failure, so MR antagonism should inhibit or delay this process [31,32,33]. Indeed, finerenone has been shown to repress increased Sgk1 levels in a murine model of CKD progression in T2D [34]. 

In addition to effects on volume distribution and pressure due to natriuresis, sodium retention in skin reservoirs is also hypothesized to be a trigger for inflammation [35]. Sodium ions can accumulate in skin reservoirs, a process that increases with age, in addition to the onset of inflammation [35]. The pro-inflammatory activity of immune cells is favored in an environment with readily available sodium while anti-inflammatory capacity is reduced [35]. This suggests that immunity is regulated by sodium availability and, consequently, that reducing local sodium, e.g., via the blockade of the MR, may be a possible remedy for autoimmune and cardiovascular diseases [35]. In support of this hypothesis, higher plasma interleukin (IL)-6 and high-sensitivity C-reactive protein levels were detected in patients undergoing peritoneal dialysis and hemodialysis, which correlated with increased muscle and skin sodium content [36].

### 2.2. Oxidative Stress—ROS Generation

MR overactivation in preclinical models increases oxidative stress in multiple cell types via increased levels of nicotinamide adenine dinucleotide phosphate oxidase (NOX) [37]. NOX controls the production of superoxide radicals in renal cells and plays a major role in ROS generation in cardiac and vascular tissue [38]. Oxidative stress also activates the nuclear factor kappa B pathway, leading to inflammation and fibrosis [39]. The further induction of MR signaling potentiates pro-inflammatory cytokine levels, resulting in the amplification of inflammation that directly increases fibrosis [40].

#### 2.2.1. Kidney ROS

In the kidneys, MR overactivation increases the availability of ROS by upregulating NOX; superoxide radicals induce malfunction in the renal vasculature and tubules while hydrogen peroxide also causes preglomerular dysfunction [37,41,42]. Increased oxidant damage and reduced nitric oxide (NO) bioavailability are also associated with ischemia in renal ischemia–reperfusion (IR) injury leading to acute kidney injury (AKI) [43]. The genetic deletion of MR in SMCs or the pharmacologic use of finerenone reduces oxidative stress production [25], and the expression of markers of tubular injury in the kidney, kidney injury molecule 1 (KIM-1), and neutrophil gelatinase-associated lipocalin (NGAL), was found to be blocked by finerenone in mice [25] and rats [26] (Table 1). Finerenone was also shown to normalize pathophysiologic increases in the oxidative stress markers malondialdehyde and 8-hydroxyguanosine after renal IR injury [26].

#### 2.2.2. Cardiac ROS

Preclinical models have evaluated ROS in the heart together with the effect of MR antagonism [70,71]. In a mouse model of cardiac fibrosis induced by short-term isoproterenol injection, finerenone reduced cardiac NOx2 expression [70] (Table 2). Another model that assessed cardiac dysfunction related to the metabolic syndrome in rats (Zucker fa/fa) showed that short-term treatment with finerenone increased myocardial tissue perfusion, reduced the level of myocardial ROS, and increased NO bioavailability [71]. Long-term effects of finerenone in a metabolic syndrome model, including modifications in myocardial structure, were also apparent [71].

#### 2.2.3. Vascular ROS

MR-dependent ROS production impedes vascular homeostasis by interrupting the differentiation and migration of bone marrow-derived endothelial progenitor cells [2]. Endothelial dysfunction due to oxidative stress is one potential mechanism for the development of cardiorenal disease that was investigated in a rat model of CKD [81]. Finerenone treatment improved endothelial dysfunction in this albuminuric CKD model by increasing NO bioavailability and superoxide dismutase protein levels. A corresponding reduction in albuminuria was also observed [81]. In the same model, finerenone treatment also dampened plasma matrix metalloproteinase-2 (MMP-2) and MMP-9 activities, lowering arterial stiffness and oxidative stress [52] (Table 1).

### 2.3. Inflammation

MR activation in cells of the immune system has been shown to drive systemic and local inflammation, organ fibrosis, and vascular, cardiac, and renal damage [82]. Cardiac damage activates an inflammatory reaction, thus generating further pro-inflammatory cytokines, including tumor necrosis factor-α (TNF-α), IL-1β, and IL-6. This transcriptional upregulation can also be triggered by increased ROS [83]. In a broad range of preclinical models, finerenone has been found to have a blocking effect on the transcriptional expression of several pro-inflammatory genes expressed in the kidney, heart, and other organs.

#### 2.3.1. Renal Inflammation

MR signaling in myeloid cells was shown to contribute to the progression of renal injury in a murine knockout model of glomerulonephritis [84]. The knockout of the MR on myeloid cells appeared to protect from renal injury, predominantly resulting from a decrease in macrophage and neutrophil recruitment [84]. This reduction in leukocytes correlated with the downregulated gene expression of proinflammatory markers, including TNF-α, inducible NO synthase, chemokine (C-C motif) ligand 2, and MMP-12 [84]. After an ischemic episode in the kidney, macrophage recruitment plays an essential role during the injury and repair phases [85]. MR activation in monocytes polarizes macrophages toward an “inflammatory M1”-like phenotype [86]. MR inhibition through finerenone promotes increased IL-4 receptor expression in murine kidney IR models and activation in the kidney and in isolated macrophages, thereby facilitating macrophage polarization to an M2 phenotype, which supports the rationale behind using MRAs to block progression of AKI into CKD [61]. Finerenone was shown to decrease the macrophage messenger RNA (mRNA) expression of proinflammatory cytokine TNF-α and M1 macrophage marker IL-1β [61]. In finerenone-treated uninephrectomized deoxycorticosterone acetate (DOCA)-treated mice, kidney retinoid-related orphan receptor (ROR) gamma t-positive T-cells were downregulated, which was accompanied by a significant reduction in the urine albumin-to-creatinine ratio (UACR), demonstrating significant renal protection [87]. In humans, IL-6 expression is induced through an MR-dependent process promoted by angiotensin II, a proinflammatory effect that can be blocked by MR antagonism [88]. Inflammation is an important precursor to fibrosis and a player in the development of AKI-induced CKD [89]. The levels of the proinflammatory cytokines IL-6 and IL-1ß were found to have increased after the induction of IR injury in untreated mice. However, this inflammatory response was prevented by finerenone administration, suggesting that MR antagonism by finerenone can modify inflammation [61]. Finerenone has also been shown to reduce the expression of renal NGAL [25,26], which is released from neutrophils during systemic inflammation and from renal tubular cells in response to tubular injury [25,26], and the pro-inflammatory cytokine monocyte chemoattractant protein-1 (MCP-1) in the DOCA-salt model of cardiorenal end-organ damage [46]. Both NGAL and MCP-1 are implicated in CKD progression in humans [48,55]. Finerenone also reduced renal osteopontin (OPN) expression in a DOCA-salt rat CKD model [46]. During renal fibrogenesis, this cytokine is thought to modulate fibroblast proliferation, macrophage activation and infiltration, cytokine secretion, and the synthesis of ECM. A previous investigation demonstrated that OPN is implicated in CKD progression, and its plasma levels are elevated from the early stages of CKD [60]. In a murine model of CKD progression in T2D (uninephrectomized mice with T2D fed a high-salt diet), finerenone offered protection from podocyte injury by reducing the expression of fibronectin, as well as inflammatory markers including MCP-1 and plasminogen activator inhibitor-1 (PAI-1), in glomeruli [34].

#### 2.3.2. Cardiac Inflammation

The role of the MR in cardiac inflammation was investigated in mice that were genetically modified to lack MR expression in cardiomyocytes. In this study, a central role for the MR was established in the initiation and progression of cardiac tissue inflammation and remodeling following induction with DOCA-salt [90]. The early innate inflammatory response was lost and the full inflammatory response was blocked, as evidenced by a reduced number of monocytes and macrophages in cardiac tissue of mice that lacked the MR on cardiomyocytes [90]. MR-deleted macrophages have an M2-type profile [91] associated with anti-inflammatory and repair properties in cardiac tissues [92]. Cardiac macrophage infiltration was also significantly blocked by finerenone in an isoproterenol-induced model of inflammation and fibrosis in mice [70]. Finerenone also reduced the cardiac mRNA expression of galectin-3 in this model [70], which is notable as it has been identified as a novel biomarker that may be associated with disease progression in patients with CKD [74]. Finally, the effect of finerenone on the previously identified inflammatory MR target NGAL (or lipocalin 2 (LCN2)) was investigated in vitro and in vivo [78]. Finerenone blunted the aldosterone-induced NGAL protein synthesis in human cardiac fibroblasts, as well as in cardiac tissue from post-MI mice. NGAL seems to play a key role in the development of cardiac dysfunction post-MI since an increase in serum NGAL levels during follow-up was significantly associated with lower 6-month left ventricular ejection fraction (LVEF) recovery in a cohort of 119 post-MI patients [78] (Table 2).

### 2.4. Fibrosis

Sustained and prolonged inflammation leads to fibrosis, an excessive accumulation of ECM and increased collagen synthesis in response to tissue injury [11,93]. This close association with prolonged inflammation generally leads to damage in a variety of organs, including the heart and the kidneys [93,94]. MR overactivation by aldosterone is thought to increase fibrosis by driving collagen expression, as well as by causing the upregulation of PAI-1, which inhibits the production of plasmin, enables the accumulation of ECM, and promotes fibrosis.

#### 2.4.1. Renal Fibrosis

In the kidneys, the development of fibrosis contributes to CKD and renal failure via the disruption of the renal tubules and surrounding blood vessels. Research in patients with kidney disease has revealed that the pro-fibrotic cytokine transforming growth factor-β (TGF-β), MCP-1, and MMP-2 are potential biomarkers for the development of fibrosis and correlate with worsening renal function [51]. Plasma PAI-1 also had a moderate correlation with fibrosis on biopsy [51]. Several preclinical models have been used to evaluate the role of the MR in the development of fibrosis and the progression of CKD and to determine the efficacy of finerenone in reducing renal fibrosis. In the DOCA-salt model of CKD in rats, finerenone reduced renal mRNA expression of the pro-fibrotic marker PAI-1 as well as renal fibrosis determined by histopathology [46]. Finerenone also reduced renal fibrosis and the renal expression of pro-fibrotic collagen type I α 1 chain (COL1A1) in a hypertensive cardiorenal rat model [64]. Furthermore, in a mouse model of renal fibrosis, finerenone dose-dependently lowered pathologic myofibroblast accumulation and collagen deposition independently of systemic blood pressure or changes in inflammatory markers [57]. Corresponding decreases in the expression of the fibrotic markers PAI-1 and naked cuticle homolog 2 (NKD2) were also observed in the kidneys [57]. NKD2 was recently identified as a myofibroblast-specific marker in human renal fibrosis [58]. In a chronic CKD rat model with renal dysfunction, increased proteinuria, and extensive tubule-interstitial fibrosis, finerenone was found to limit renal collagen deposition and fibrosis, as scored by histopathology [26]. Finerenone administration prevented an increase in the renal expression of the pro-fibrotic cytokine TGF-β and collagen-I [26]. Similarly, in a mouse CKD model of unilateral, IR-induced tubulo-interstitial fibrosis, finerenone significantly reduced the severity of renal fibrosis [61] (Table 1).

#### 2.4.2. Cardiac and Vascular Fibrosis 

Pathophysiologic MR overactivation promotes cardiac and vascular fibrosis with and without concomitant oxidative damage and inflammation [27,72]. The effects of finerenone on cardiac and vascular fibrosis have been investigated in several preclinical models. Finerenone was shown to reduce cardiac fibrosis in a hypertensive cardiorenal rat model and the cardiac expression of pro-fibrotic PAI-1 in mouse models of renal fibrosis [57,64]. The treatment of a genetic mouse model of Duchenne muscular dystrophy (DMD) with finerenone also prevented significant reductions in myocardial strain rate, the earliest sign of human DMD cardiomyopathy, with a corresponding reduction in the accumulation of fibrotic tissue in the heart [95]. In a transgenic mouse model with the cardiac-specific overexpression of Rac1 (RacET), a model of left ventricular and left atrial fibrosis, 5 months of treatment with finerenone significantly reduced the cardiac mRNA expression of TGF-β; myocardial fibrosis was also reduced with finerenone [72]. In an isoproterenol-induced cardiac fibrosis mouse model, the examination of cardiac tissues revealed an increased cardiac collagen accumulation. Treatment with finerenone significantly reduced the isoproterenol-induced pro-fibrotic effect and ameliorated the isoproterenol-induced increase in tenascin-X, a protein involved in the regulation of collagen deposition and degradation [70]. Furthermore, the expression of classical fibrotic molecules (including TGF-β, COL1A1, and galectin-3) was increased through isoproterenol treatment, the effects of which were substantially reduced with finerenone treatment [70] (Table 2).

The role of the MR in the development of atrial fibrosis, a predisposing factor for the development of atrial fibrillation (AF) has also been studied. The left atrial myocardium of patients with AF exhibited an increased hydroxyproline content, a marker of fibrosis, compared with patients in sinus rhythm. MR antagonism by finerenone prevented the aldosterone-induced upregulation of connective tissue growth factor (CTGF) protein expression (a marker for structural remodeling) and cardiac fibrotic remodeling, as well as lysyl oxidase (which is involved in collagen cross-linking) [72]. In a mouse model of post-MI-induced heart failure, treatment with finerenone for 2 months improved left ventricular compliance and elastance, as well as reducing interstitial fibrosis [96]. 

### 2.5. Hypertrophy/Remodeling

Cardiac hypertrophy occurs as a compensatory response to a sustained increased in stress enacted on the left ventricular wall [97]. Cardiac hypertrophy can become pathologic and contribute to the development of HF. MR blockade has been shown to suppress cardiac hypertrophy and remodeling in animal models of pressure overload [98] and the knockout of the MR, specifically in myeloid cells, attenuated cardiac hypertrophy following cardiac and vascular damage [91]. In a transverse aortic constriction model in mice, finerenone reduced the cardiac gene expression of troponin T type 2, leading to a significant reduction in left ventricular wall thickening [80]. Finerenone was also able to reduce cardiac hypertrophy and renal damage in DOCA-salt-treated rats [46]. Finerenone has also been shown to significantly reduce the apoptosis of ECs and simultaneously attenuate SMC proliferation, resulting in accelerated endothelial healing and the reduced neointima formation of injured vessels following electric injury of the murine carotid artery [99]. Thus, finerenone appears to provide favorable vascular effects through restoring vascular integrity and preventing adverse vascular remodeling [99].

## 3. Beyond the Cardiovascular and Renal Systems: Consequences of MR Overactivation in Other Organs

As outlined earlier in this article, end-organ damage and related dysfunction of the heart, kidneys, and vasculature can be the consequences of pathophysiological mechanisms including oxidative stress and ROS generation, ECM remodeling and hypertrophy, and inflammation and fibrosis. Interestingly, these disease drivers are also found in several other organ systems where the role of MR overactivation is less well-established. An example of tissue damage based on long-term inflammatory processes with a high prevalence in the aged male population is benign prostatic hyperplasia (BPH). Recently, the role of chronic inflammation in BPH was investigated [100], and a drug combination with anti-inflammatory and antioxidant activity was proven able to reduce prostatic inflammation in vitro [101]. However, a respective role of MR overactivation in prostate disease remains ambiguous since neither MR nor 11βHSD2 could be detected in an immunolocalization study using nonpathological human prostate tissue from autopsy specimens [102]. Nevertheless, future investigations might show whether MR overactivation plays a role in human prostate pathology, as well as in other conditions that involve the same pathophysiological drivers of disease.

### 3.1. Lung

Cardiac and pulmonary fibrosis share several common signaling pathways, including TGF-β-related collagen synthesis [72,103,104]. However, the role of MR overactivation in the development of pulmonary fibrosis is less well-understood. Two preclinical models of idiopathic pulmonary fibrosis have shown that finerenone demonstrates antifibrotic and anti-inflammatory activity via a reduction in the levels of IL-6, a pro-fibrotic and proinflammatory cytokine, in the lungs [105]. Finerenone also reduced the upregulation of the proinflammatory cytokines TNF-α and IL-1β, in addition to lowering the percentage of COL1A1-positive lung tissue [105] (Table 3). Furthermore, clinical data from FIDELITY support a potential role for MR antagonism in the prevention of lung inflammation, as the incidence of pneumonia was lower with finerenone vs. placebo in patients with CKD and T2D [22]. MR expression is increased during human pulmonary arterial hypertension in humans, and finerenone was found to impair remodeling in the vasculature and pulmonary hypertension via MR blockade through the minimization of the buildup of pulmonary artery SMCs in monocrotaline and Sugen/Hypoxia rat models of pulmonary arterial hypertension [106]. Transgenic mice overexpressing MR had a higher right ventricular systolic pressure and hypertrophy in the right ventricle, in addition to the remodeling of small pulmonary vessels—effects that were reduced by treatment with finerenone [106].

### 3.2. Adipose Tissue

There are three main types of adipocytes in adipose tissue with differing functions: white, beige, and brown. White adipocytes are most common and function as an energy store. Brown adipocytes are metabolically active and are involved in expending energy and producing heat, whereas beige adipocytes have thermogenic potential [113]. The browning of white adipose tissue into beige has a favorable effect on whole body metabolism, as brown and beige adipocytes have the potential to counteract metabolic diseases such as T2D [114]. Aldosterone promotes adipose tissue expansion via MR activation [110]. Visceral adipose tissue is also thought to increase aldosterone levels via the release of adipokines and hence serves as another source of aldosterone in obese patients [115]. MRAs have been previously shown to reduce white adipose tissue expansion, stimulate the browning of this tissue, and activate interscapular brown adipose tissue. In a mouse model of obesity, finerenone increased the recruitment of brown adipocytes, which may have contributed to the reduction in insulin resistance that was observed following finerenone treatment [110]. Finerenone also increased the mRNA expression of molecules involved in thermogenesis such as uncoupling protein-1 (UCP-1), peroxisome proliferator-activated receptor-γ (PPAR-γ) coactivator 1 alpha, and beta-3 adrenoreceptor in brown adipose tissue, suggesting that finerenone may be a promising pharmacologic agent to treat human metabolic diseases associated with adipose tissue dysfunctions [110] (Table 3).

### 3.3. Eye

The MR is expressed in various retinal cell populations of the eye including vascular cells (ECs and pericytes), ganglion cells, macroglial Müller cells, microglia, and retinal pigmented epithelial cells [116,117,118,119]. In Müller cells, aldosterone and salt stimulate the expression of epithelial sodium channels, the transmembrane water channel aquaporin 4, and the inwardly rectifying potassium channel Kir4.1 [118,120]. The activation of the MR by aldosterone has also been demonstrated to contribute to choroidal vasodilation, subretinal fluid accumulation, retinal inflammation, and oxidative stress [121,122]. A rat model of oxygen-induced retinopathy has been used to investigate the function of the MR and the impact of MR antagonism in the eye [117]. Aldosterone was shown to increase retinal EC proliferation and tubule formation, which were reduced by systemic MR antagonism [117]. Additionally, MR antagonism attenuated aldosterone-induced inflammation, with a reduction in leukostasis and MCP-1 levels being observed [117]. The expression of retinal MR was found to be increased in diabetic patients compared with nondiabetic controls [123], but the role of the MR in the development of diabetic retinopathy has yet to be fully elucidated. Intraocular treatment with spironolactone decreased retinal edema in Goto-Kakizaki rats with T2D by preventing retinal barrier breakdown [123]. These rats also showed an increased expression of the inflammatory markers intercellular adhesion molecule 1 (ICAM-1) and MCP-1, which was also reversed with MR antagonism [123]. Recently, two sub-studies of patients with CKD and T2D from the FIDELITY population assessed whether finerenone is able to delay the progression of diabetic retinopathy. After 2 years of treatment, a lower proportion of patients had experienced a vision-threatening complication with finerenone vs. placebo (5/134 (3.7%) vs. 7/110 (6.4%), respectively) and fewer patients required ocular interventions with finerenone [124]. 

Figure 1 summarizes the components of pathophysiologic MR overactivation, which are counteracted by finerenone in different organs and cell types, as well as relevant genes and biomarkers, which are modulated by finerenone in respective preclinical models.

## 4. Finerenone in Clinical Studies: Mechanistic Explanations and Future Prospects

### 4.1. Renal and Cardiovascular Outcomes in FIGARO/FIDELIO and Mechanistic Explanations

The FIDELIO-DKD and FIGARO-DKD trials, as well as FIDELITY, demonstrated a beneficial effect of MR antagonism by finerenone on renal and cardiovascular outcomes across patients with CKD and T2D [20,21,22]. An initial drop in estimated GFR (eGFR) was observed in the finerenone arm as early as 1 month [20]. Similar short-term drops in eGFR have been observed with sodium–glucose cotransporter-2 inhibitors (SGLT-2is) and renin–angiotensin system inhibitors, and they are widely believed to be hemodynamic in nature [125], correlating to a certain extent with a reduction in blood pressure but also consistent with slower declines in renal function [126]. Accordingly, in the FIDELIO-DKD study, finerenone treatment resulted in a modest reduction in office blood pressure (mean change in systolic blood pressure from baseline to month 12: −2.1 mmHg with finerenone vs. 0.9 mmHg with placebo) [20] that was responsible for a small proportion of the clinical effect of finerenone in both the kidneys and the heart [127]. By 4 months of treatment, a significant reduction in albuminuria (UACR) was observed with finerenone vs. placebo (31–32% lower with finerenone) [20,22,128]. Notably, this was the first measurement of UACR taken in the FIDELIO-DKD and FIGARO-DKD studies after the initiation of treatment. However, in the ARTS phase II study, a reduction in UACR with finerenone was evident after just over 2 weeks of treatment [129]. This early reduction in albuminuria, together with the initial eGFR drop, suggests that some rapid benefits of finerenone may be partly mediated by natriuretic and renal hemodynamic mechanisms in patients with CKD and T2D. Early benefits of finerenone on clinical outcomes were also revealed by the separation of the Kaplan–Meier curves at 6 months for the composite cardiovascular outcome of time to cardiovascular death, nonfatal MI, nonfatal stroke, and hospitalization for HF (HHF) in the FIDELIO-DKD study [20]. The cardiovascular benefits of finerenone were mainly driven by a reduction in the risk of cardiovascular death and HHF. These observations support the notion that some of the benefits of finerenone may be initially mediated by interfering with sodium retention and endothelial dysfunction [130]. The amelioration of volume retention with finerenone because of its effects on sodium retention, combined with an improvement in endothelial dysfunction via interfering with ROS generation, are hypothesized to have contributed to these findings [81,130].

In addition to these early effects, results from FIDELIO-DKD and FIGARO-DKD also illustrated a longer-term action of finerenone, hypothesized to be due to a reduction in MR-mediated inflammation and fibrosis [19]. These benefits have been observed as a reduced rate of long-term eGFR decline vs. placebo, with the intersection of the least-squares mean change from baseline in the eGFR slopes at 28 months in FIDELIO-DKD and 36 months in FIGARO-DKD [20,128]. Effects of finerenone on clinical renal outcomes are also slower to develop, as seen in the later divergence around 20–24 months of the Kaplan–Meier curves for the composite kidney outcomes of time to end-stage kidney disease, sustained ≥40% or 57% decreases in eGFR from baseline, or renal death [20,21,22]. However, this observation is also related to the lower event rates of renal outcomes in the beginning of the trials. A continued long-term improvement on cardiovascular outcomes has been observed, and it has been hypothesized to be due in part to the effect of finerenone on vascular stiffness [130]. Furthermore, benefits of finerenone on the incidence of new-onset AF were demonstrated in FIDELIO-DKD; these benefits were noticed following the separation of the Kaplan–Meier curves at month 6 and continued for the duration of the study [131]. Furthermore, a significant reduction in HHF in patients with left ventricular hypertrophy at baseline was apparent within 6 months of treatment and also continued for the duration of the study [132]. One possible mechanism for these benefits is an inhibition of aldosterone-mediated atrial remodeling associated with CKD and T2D [131]. The basis of the cardiovascular outcomes with finerenone likely stems from its immediate therapeutic effects, including the inhibition of sodium retention, modest effects on (local) hemodynamics, and protection from vascular endothelial dysfunction induced by ROS. A reduction in ROS generation with finerenone may also be a contributing factor to the inflammatory processes and the reduction in albuminuria. The anti-fibrotic and anti-hypertrophic activity of finerenone may be long-term consequences of accumulating structural healing. The time-dependent modulation of these mechanistic components as a consequence of MR overactivation, together with the effect of finerenone on the clinical parameters eGFR and UACR and on the clinical cardiovascular and renal composite outcomes, are summarized in Figure 2.

### 4.2. “Systemic” MR Antagonism vs. “Local” (i.e., Proximal Tubule-Specific) SGLT-2 Inhibition and Potential for Combination Therapy

SGLT-2 is selectively expressed in the human kidney, predominantly at the apical membrane of renal proximal convoluted tubules [133], where it is employed in reabsorbing 80–90% of glucose into the blood [134]. SGLT-2is mitigate against the pathologic hyper-reabsorption of sodium in the early proximal tubule, which is especially beneficial to patients with diabetes because the rise in filtered glucose enhances SGLT-2 activity [135]. Impeded sodium flow restores hyperfiltration in the glomerulus and the vasodilation of the afferent arteries via tubuloglomerular feedback, ameliorating renal damage and long-term renal function [135]. SGLT-2is also cause weight loss, reduce systemic blood pressure, increase ketone generation, and heighten insulin sensitivity [136]. However, these effects are only described in people who have reasonable renal function, i.e., eGFR > 45 mL/min/1.73 m^2^. Nonetheless, SGLT-2is have also demonstrated cardiorenal benefits in patients with an eGFR level of <45 mL/min/1.73 m^2^ [137,138], suggesting mechanisms other than glucose or sodium reduction.

In contrast to SGLT-2, the MR is broadly expressed across multiple organ systems and in numerous tissues. This review clearly indicates the systemic effect of MR antagonism and subsequent potential to be utilized in benefiting multiple organ systems. It also demonstrates that while SGLT-2is can be used to treat patients with HF or CKD with or without T2D, they remain constrained by a mechanism predominantly limited to the kidneys. The distinct modes of action of SGLT-2i and finerenone suggests that the two classes of medication could be independently combined. A preclinical rat model of hypertension-induced end-organ damage was used to assess the combination of finerenone with an SGLT-2i [64]. Cardiovascular and renal protection in the model were pronounced with combination therapy compared with either agent alone. The complementary metabolic and hemodynamic effects distinctly reduced morbidity and mortality [64]. Combination therapy with finerenone and an SGLTI-2i has also been assessed in patients with CKD and T2D. In FIDELIO-DKD, treatment with finerenone reduced albuminuria with a consistent effect on cardiorenal outcomes in patients who were already receiving an SGLT-2i at baseline [139]. Similarly, an analysis of patients in FIDELITY who received concomitant treatment with finerenone and an SGLT-2i confirmed these findings. Data from the FIDELITY analysis also suggest that the use of an SGLT-2i may offer protection from hyperkalemia events when used in combination with finerenone [140]. The authors of other studies have investigated the combination of an SGLT-2i with a steroidal MRA. A small study of dapagliflozin with eplerenone in patients with CKD showed a robust additive UACR-lowering effect [141]. In HFrEF, the use of an MRA in combination with an SGLT-2i (dapagliflozin in Dapagliflozin and Prevention of Adverse Outcomes in Heart Failure (DAPA-HF) and empagliflozin in Empagliflozin Outcome Trial in Patients with Chronic Heart Failure and Reduced Ejection Fraction (EMPEROR-Reduced)) did not alter the cardiorenal benefit offered by treatment with SGLT-2i alone [142,143].

### 4.3. Potential Future Clinical Indications for Finerenone and Ongoing Clinical Studies

Clinical trials with finerenone in new therapy areas and existing indications are ongoing (Table 4). Regarding the existing indication of CKD and T2D, an observational study of treatment patterns in routine medical care and patient characteristics (a noninterventional study providing insights into the use of finerenone in a routine clinical setting (FINE-REAL)) is ongoing. Another trial is underway to assess whether the effect of combination therapy with an SGLT-2i observed in preclinical models can translate into clinical benefits for patients. The COmbinatioN effect of FInerenone anD EmpaglifloziN in participants with CKD and T2D using an UACR Endpoint (CONFIDENCE) study will investigate the efficacy and safety of finerenone in combination with empagliflozin compared with either empagliflozin alone or finerenone alone in patients with CKD and T2D. The primary outcome measure is change in albuminuria from baseline measured by UACR. Two trials have also been initiated in patients with nondiabetic CKD. The FInerenone for the treatment of children with chrOnic kidNey disease and proteinuriA (FIONA) study has been designed to investigate finerenone in addition to the standard of care in pediatric patients with CKD with a primary outcome focused on a percent change in urine protein to creatinine ratio from baseline. A randomized, double-blind, placebo-controlled, parallel-group, multicenter phase III study to investigate the efficacy and safety of finerenone, in addition to standard of care, on the progression of kidney disease in patients with nondiabetic chronic kidney disease (FIND-CKD) will assess the change in renal function measured by eGFR with finerenone vs. placebo in adults with nondiabetic CKD. Finally, the phase III FINerenone trial to investigate Efficacy and sAfety superioR to placebo in paTientS with Heart Failure (FINEARTS-HF) will evaluate the efficacy and safety of finerenone in patients with symptomatic HF (New York Heart Association class II–IV) and left ventricular ejection fraction ≥40% who are ambulatory or primarily hospitalized for HF. The primary outcome will measure the number of cardiovascular deaths and HF events. This trial will build on the evidence gained on finerenone from the ARTS-HF trial in patients with worsening chronic HFrEF (left ventricular ejection fraction of ≤40%) [144].

## 5. Conclusions

We have provided a significant body of preclinical evidence to support the beneficial effects of the nonsteroidal MRA finerenone in the cardiorenal system (in terms of the modulation of oxidative stress, inflammation, and fibrosis) to lower the risk of adverse cardiovascular and renal outcomes in experimental models. A wide range of biomarkers that are modulated by finerenone have been identified throughout the course of this research. It is clear that the overactivation of the MR plays a key role in the pathophysiology of renal and cardiovascular disease and that finerenone is able to inhibit these processes in animal models of CKD and cardiac dysfunction. Indeed, two large phase III trials (FIDELIO-DKD and FIGARO-DKD) have confirmed the cardiorenal benefits of finerenone in patients with CKD and T2D, respectively. A small proportion of the clinical effect of finerenone has been attributed to a modest reduction in blood pressure observed in these trials, with the remaining mechanisms yet to be fully elucidated. Finerenone has demonstrated a rapid effect on renal hemodynamics, as well as early improvements in CV outcomes, suggesting that benefits may be initially mediated by interfering with sodium retention and endothelial dysfunction. A longer-term action of finerenone, as observed in FIDELIO-DKD and FIGARO-DKD, is postulated to be due to a reduction in MR-mediated inflammation and fibrosis.

We have also reported here that finerenone has demonstrated preclinical activity in other organs beyond the cardiovascular and renal systems, suggesting that finerenone may have potential to offer therapeutic benefits in other clinical indications in the future such as for the treatment of idiopathic pulmonary fibrosis, metabolic diseases, and diabetic retinopathy. Ongoing clinical studies with finerenone will expand the knowledge within the existing indications of CKD and T2D by identifying treatment patterns in this patient population, and a dedicated trial will assess whether the combination of finerenone and an SGLT-2i can provide additional cardiorenal protection compared with either agent alone. Trials are also in progress to assess the efficacy of finerenone in new indications including adult and pediatric patients with nondiabetic CKD, as well as in patients with symptomatic HF and LVEF ≥ 40%.

In summary, finerenone has demonstrated efficacy in terms of cardiorenal outcomes in preclinical models of CKD and cardiac dysfunction, as well as in clinical studies of patients with CKD and T2D. Given the expression of the MR in a wide range of organ systems throughout the body, finerenone may have the potential to provide therapeutic benefit for diseases beyond the cardiovascular and renal systems in the future.

## Figures and Tables

**Figure 1 ijms-23-09243-f001:**
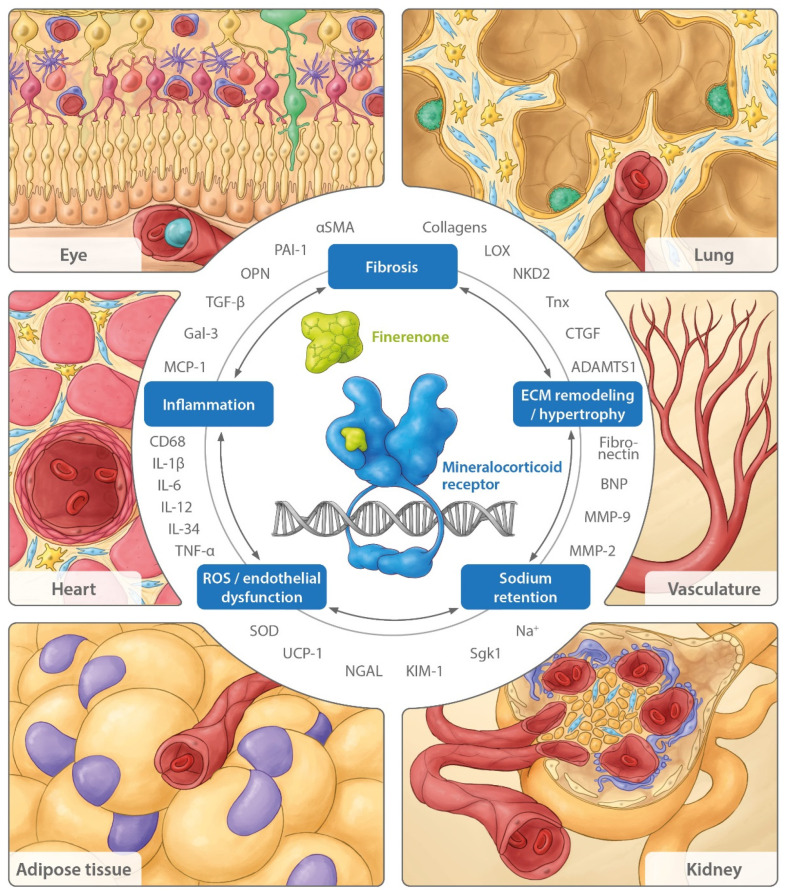
Components of pathophysiological MR overactivation that are counteracted by finerenone in different organs and cell types, including relevant biomarkers. Antagonism by finerenone counteracts the pathophysiological components of MR overactivation including sodium retention, ROS generation and endothelial dysfunction, inflammation, fibrosis, ECM remodeling and hypertrophy. Biomarkers with a modified expression induced by MR antagonism with finerenone in diverse preclinical disease models are depicted around the mechanistic components. Organs including critical functional units (e.g., retina in the eye, alveolus in the lung, and glomerulus in the kidney) and relevant specific cell types (e.g., fibroblasts in light blue and macrophages in yellow) with a documented MR-based pathophysiology, as described in the text, are framing the view. αSMA, alpha smooth muscle actin; ADAMTS1, a disintegrin and metalloproteinase with thrombospondin type 1 motif 1; BNP, B-type natriuretic peptide; CD68, cluster of differentiation 68; CTGF, connective tissue growth factor; ECM, extracellular matrix; Gal-3, galectin-3; IL, interleukin; KIM-1, kidney injury molecule 1; LOX, lysyl oxidase; MCP-1, monocyte chemoattractant protein-1; MMP, matrix metalloproteinase; NGAL, neutrophil gelatinase-associated lipocalin; NKD2, naked cuticle homolog 2; OPN, osteopontin; PAI-1, plasminogen activator inhibitor-1; ROS, reactive oxygen species; Sgk1, serum- and glucocorticoid-regulated kinase 1; SOD, superoxide dismutase; TGF-β, transforming growth factor-β; TNF-α, tumor necrosis factor-α; Tnx, tenascin-X; UCP-1, uncoupling protein-1.

**Figure 2 ijms-23-09243-f002:**
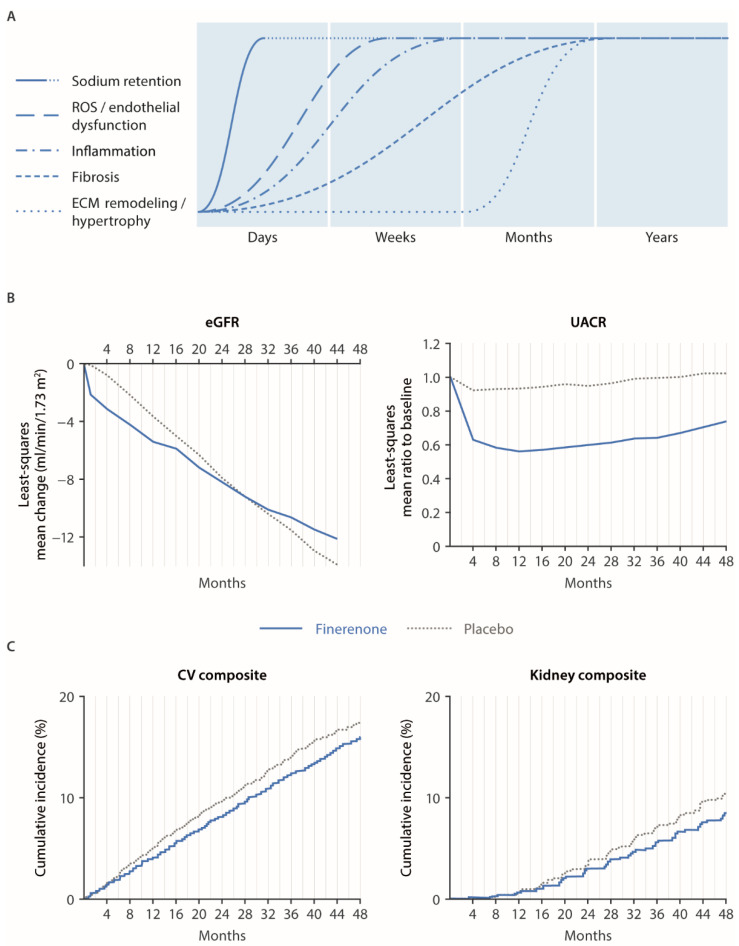
Onset of pathophysiological mechanisms of mineralocorticoid receptor overactivation and onset of clinical effects with finerenone in the phase III outcome program in patients with CKD and T2D. (**A**) Onset of the individual mechanistic components of mineralocorticoid receptor overactivation, i.e., sodium retention, ROS generation, inflammation, fibrosis, and hypertrophy/remodeling, are displayed over time (approximate times). (**B**,**C**) Onset of clinical effects with finerenone (blue lines) vs. placebo (dotted grey lines) as determined in the large phase III clinical outcome program (FIDELIO/FIDELITY). (**B**) eGFR slopes from placebo and finerenone groups as determined in FIDELIO (adapted from Bakris et al. [20]) and respective UACR curves as determined in FIDELITY (adapted from Agarwal et al. [22]). (**C**) CV composite outcome (time to CV death, nonfatal myocardial infarction, nonfatal stroke, or hospitalization for heart failure) and renal composite outcome (time to renal failure, sustained ≥57% decrease in eGFR from baseline, or renal death) in FIDELITY (adapted from Agarwal et al. [22]). CV, cardiovascular; ECM, extracellular matrix; eGFR, estimated glomerular filtration rate; ROS, reactive oxygen species; UACR, urine albumin-to-creatinine ratio.

**Table 1 ijms-23-09243-t001:** Renal biomarkers modulated by finerenone in preclinical studies and their clinical association.

Marker	Effect of Finerenone in Preclinical Models	Function/Role	Evidence for Clinical Association
Kidney			
Fibronectin	↓ Kidney mRNA expression and protein levels in model of CKD progression in T2D [34]	Glycoprotein in the glomerular mesangial ECM	CKD progression [44]
KIM-1	↓ Kidney expression in rat model of AKI [43]	Kidney injury molecule-1 (marker of tubule cell injury)	Acute kidney injury [45]
MCP-1 (CCL-2)	↓ Kidney mRNA expression in DOCA-salt model of CKD [46] and in model of CKD progression in T2D [34]	Pro-inflammatory cytokine (regulating monocyte/macrophage recruitment)	CKD/CKD progression [47,48,49,50,51]
MMP-2	↓ Kidney mRNA expression in DOCA-salt model of CKD [46] ↓ Plasma activities in nondiabetic CKD model [52]	ECM homeostasis	CKD/CKD progression [51,53]
MMP-9	↓ Plasma activities in nondiabetic CKD model [52]	ECM homeostasis	CKD/CKD progression [53,54]
NGAL (LCN2)	↓ Kidney mRNA expression [25,26]	Involved in innate immunity and in response to tubular injury	CKD/CKD progression [55,56]
NKD2	↓ Kidney expression in mouse models of kidney fibrosis [57]	Pro-fibrotic cytokine	Kidney fibrosis [58]
OPN (=Spp1)	↓ Kidney mRNA expression in DOCA-salt models of CKD [46,59]	Pro-inflammatory cytokine involved in chronic inflammation Key cytokine-regulating tissue repair, promoting collagen organization and regulating ECM and myofibroblast interactions	CKD/CKD progression [60]
PAI-1	↓ Kidney mRNA expression in DOCA-salt model of CKD [46], two models of kidney fibrosis [57] and model of CKD progression in T2D [34] but ↑ mRNA expression in macrophages from kidney tissue in IR injury model [61]	Serine protease inhibitor, which limits fibrinolysis; marker of inflammation and remodeling	CKD progression [62], kidney fibrosis [51]
Sgk1	↓ Kidney mRNA expression and protein levels in model of CKD progression in T2D [34]	Promotes inflammation and fibrosis	-
TGF-β	↓ Kidney mRNA expression in model of AKI-mediated CKD [26]	Pro-fibrotic cytokine	CKD/CKD progression [51,63]
COL1A1	↓ Kidney mRNA expression in model of AKI-mediated CKD [26], a nondiabetic hypertensive cardiorenal disease model [64], and two models of kidney fibrosis [57]	ECM molecule	-
E-cadherin	↑ Protein expression in model of AKI-mediated CKD [26]	Cell adhesion molecule	-
Nrf2	↑ mRNA expression in model of AKI-mediated CKD [26]	Regulator of antioxidant defense	CKD progression [65]
SOD-3	↑ Protein expression in model of AKI-mediated CKD [26]	Antioxidant enzyme	SOD-3 is depleted from human CKD kidneys [66]
Endothelin-B receptor	Prevents cysteine sulfenic acid modification of ET-B receptor in model of IR-induced AKI [26,43]	Regulator of vascular function	-
MDA	Kidney levels in model of IR-induced AKI [26]	Oxidative stress marker	-
8-OHdG	Plasma levels in model of IR-induced AKI [26]	Oxidative stress marker	-
IL-6	↓ Kidney mRNA expression in IR injury model [61]	Pro-fibrotic and pro-inflammatory cytokine	CKD progression [67]
IL-1β	↓ Kidney mRNA expression in IR injury model [61]	Pro-inflammatory cytokine and M1 macrophage marker	CKD progression [67]
TNF-α	↓ Kidney mRNA expression in IR injury model [61]	Pro-inflammatory cytokine	CKD progression [47,67,68]
Mannose receptor	↑ mRNA expression in macrophages from kidney tissue in IR injury model [61]	Anti-inflammatory marker	-
PPAR-γ	↑ mRNA expression in macrophages from kidney tissue in IR injury model [61]	Anti-inflammatory marker	-
IL-10	↑ mRNA expression in macrophages from kidney tissue in IR injury model [61]	Anti-inflammatory cytokine	-
Arginase 1	↑ mRNA expression in macrophages from kidney tissue in IR injury model [61]	Anti-inflammatory marker	CKD progression
IL-34	↓ Kidney mRNA expression in DOCA-salt model of CKD [59]	Monocyte growth and survival	Worsening of CKD and severity of renal dysfunction [69]

↑, increased; ↓, decreased; 8-OHdG, 8-hydroxy-2′-deoxyguanosine; AKI, acute kidney injury; CCL-2, C-C motif chemokine ligand 2; CKD, chronic kidney disease; COL1A1, collagen type I α 1 chain; DOCA, deoxycorticosterone acetate; ECM, extracellular matrix; ET-B, endothelin-B receptor; IL, interleukin; IR, ischemia–reperfusion; KIM-1, kidney injury molecule 1; LCN2, lipocalin 2; MCP-1, monocyte chemoattractant protein-1; MDA, malondialdehyde; MMP, matrix metalloproteinase; mRNA, messenger RNA; NGAL, neutrophil gelatinase-associated lipocalin; NKD2, naked cuticle homolog 2; Nrf2, nuclear factor erythroid-2-related factor 2; OPN, osteopontin; PAI-1, plasminogen activator inhibitor-1; PPAR-γ, peroxisome proliferator-activated receptor-γ; Sgk1, serum- and glucocorticoid-regulated kinase 1; SOD-3, superoxide dismutase-3; Spp1, secreted phosphoprotein 1; T2D, type 2 diabetes; TGF-β, transforming growth factor-β; TNF-α, tumor necrosis factor-α.

**Table 2 ijms-23-09243-t002:** Cardiac biomarkers modulated by finerenone in preclinical studies and their clinical association.

Marker	Effect of Finerenone in Preclinical Models	Function/Role	Evidence for Clinical Association
Cardiac			
CTGF	↓ Protein expression in cardiac fibroblasts [72]	Pro-fibrotic cytokine that induces collagen production and subsequent pro-fibrotic enzymes	Cardiac fibrosis and dysfunction [73]
Fibronectin	↓ Protein expression in cardiac fibroblasts [72]	Glycoprotein in fibrotic cardiac tissue	-
Galectin 3	↓ Cardiac mRNA expression after isoproterenol treatment [70]	Implicated in cardiac and renal inflammation and fibrosis	CKD progression [74,75]
COL1A1	↓ Cardiac mRNA expression after isoproterenol treatment [70]	ECM molecule	Heart failure progression [76]
LOX	↓ Protein expression in cardiac fibroblasts [72]	Downstream mediator of CTGF, important for collagen cross-linking	Cardiac fibrosis [77]
NGAL (LCN2)	↓ Protein expression in human cardiac fibroblasts and ↓ cardiac NGAL expression in mice post-MI [78]	Involved in innate immunity and cardiovascular extracellular matrix remodeling after MR activation	Serum NGAL levels were associated with lower 6-month LV ejection fraction recovery in post-MI patients [78]
Nox2	↓ Cardiac mRNA expression after isoproterenol treatment [70]	ROS-generating enzyme	Adverse myocardial remodeling in end-stage DCM [79]
TGF-β	↓ Cardiac expression [72] after isoproterenol treatment [70]	Pro-fibrotic cytokine	-
Tnnt2	↓ Cardiac mRNA expression [80]	Contractile protein	-
Tenascin-X	↓ Cardiac mRNA expression after isoproterenol treatment [70]	Pro-fibrotic cytokine	-

↓, decreased; CKD, chronic kidney disease; COL1A1, collagen type I α 1 chain; CTGF, connective tissue growth factor; DCM, dilated cardiomyopathy; ECM, extracellular matrix; LCN2, lipocalin 2; LOX, lysyl oxidase; LV, left ventricular; MI, myocardial infarction; MR, mineralocorticoid receptor; mRNA, messenger RNA; NGAL, neutrophil gelatinase-associated lipocalin; Nox2, nicotinamide adenine dinucleotide phosphate oxidase 2; ROS, reactive oxygen species; TGF-β, transforming growth factor-β; Tnnt2, troponin T type 2.

**Table 3 ijms-23-09243-t003:** Biomarkers determined in other organs that are modulated by finerenone in preclinical studies and their clinical association.

Marker	Effect of Finerenone in Preclinical Models	Function/Role	Evidence for Clinical Association
Other			
IL-10	↓ Pulmonary expression [105]	Anti-inflammatory cytokine M2 macrophage marker/wound-healing phenotype	Pulmonary fibrosis [107]
TNF-α	↓ Pulmonary expression [105]	Pro-inflammatory cytokine; involved in innate immune response	-
IL-1β	↓ Pulmonary expression [105]	Pro-inflammatory cytokine and M1 macrophage marker	Mortality in acute exacerbations of idiopathic pulmonary fibrosis [108]
COL1A1	↓ Pulmonary expression [105]	ECM molecule	Pulmonary fibrosis [109]
IL-6	↓ Pulmonary expression [105]	Pro-fibrotic and pro-inflammatory cytokine	-
IL-12	↓ Pulmonary expression [105]	Pro-fibrotic and pro-inflammatory cytokine	Pulmonary fibrosis [107]
PGC1-α	↑ Expression in interscapular brown adipose tissue [110]	Thermogenesis	-
Adrb3	↑ Expression in interscapular brown adipose tissue [110]	Thermogenesis	-
UCP-1	↑ Expression in interscapular brown adipose tissue [110]	Thermogenesis	-
SOD	↑ Vasculature expression [81]	Antioxidant	Vascular dysfunction [111]
eNOS	↑ Vasculature expression [81]	Catalyst for production of NO	Enhanced NO signaling is associated with reduced risks of coronary heart disease, peripheral arterial disease, and stroke [112]

↑, increased; ↓, decreased; Adrb3, adrenoceptor beta 3; COL1A1, collagen type I α 1 chain; ECM, extracellular matrix; eNOS, endothelial nitric oxide synthase; IL, interleukin; NO, nitric oxide; PGC1-α, peroxisome proliferator-activated receptor-gamma coactivator 1-alpha; SOD, superoxide dismutase; TNF-α, tumor necrosis factor-α; UCP-1, uncoupling protein-1.

**Table 4 ijms-23-09243-t004:** Ongoing clinical studies with finerenone.

Trial Name	NCT Number	Indication	Planned Enrollment	Primary Endpoint	Estimated Study Completion
FINE-REAL	NCT05348733	CKD and T2D	4000	Treatment patterns *	February 2026
CONFIDENCE	NCT05254002	CKD and T2D	807	Relative change in UACR from baseline to 180 days	January 2024
FIND-CKD	NCT05047263	Nondiabetic CKD	1580	Mean rate change of total eGFR slope from baseline to month 32	December 2025
FIONA	NCT05196035	Pediatric CKD	219	≥30% UPCR reduction from baseline to day 180	September 2026
FINEARTS-HF	NCT04435626	HF with LVEF ≥40%	5500	CV death and HF events	May 2024

* Including clinical characteristics, reasons for introducing/discontinuing finerenone, planned and actual duration of finerenone treatment, dose administered, frequency of treatment, and concomitant medications. CKD, chronic kidney disease; CV, cardiovascular; eGFR, estimated glomerular filtration rate; HF, heart failure; T2D, type 2 diabetes; UACR, urine albumin-to-creatinine ratio; UPCR, urine protein-to-creatinine ratio.

## Data Availability

The data used in this article are sourced from materials mentioned in the references section.

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
