# Peer review of "Nonsteroidal Mineralocorticoid Receptor Antagonism by Finerenone—Translational Aspects and Clinical Perspectives across Multiple Organ Systems"

_ijms, 2022, doi:10.3390/ijms23169243_

Round 1

Reviewer 1 Report

The authors provide an extensive and up to date review on the action of finerenone, a (new) non steroidal MR antagonist, on various organ systems where MR overactivation is known to induce deleterious responses, such as oxidative stress, inflammation and fibrosis. This review is extensively documented and well organized in preclinical and clinical data that are adequately summarized in relevant tables and figures very useful for the reader.

 I have only a few comments.

 Specific comments:

 1) In the introduction (p.2, l.64), the authors describe the various deleterious actions resulting from  MR activation in the heart. However, they do not mention the rapid (within days) remodeling of ion channels that occurs in cardiomyocytes upon stimulation with aldosterone. This remodeling leads to a positive chronotropic response of the myocyte beating frequency in vitro and to ventricular arrhythmias in vivo, which are responsible for hyperaldosteronism-related sudden death. This electrical remodeling, involving the (re)expression of low-threshold (T-type) voltage-operated calcium channels, probably contributes to the higher risk of early hospitalization for cardiac fibrillation (and death) that is prevented by MRA therapy (p.14, l.483). Although the beneficial effect was observed mostly with spironolactone and eplerenone, this mechanism of action should be also considered for finerenone.

 2) The references are not evenly cited, some are with only starting page number, others with initial and final pages. References 101, 122 and 124 are apparently incomplete.

Minor point:

 P.19, l.614: SGLT-2i (not SLGT)

Author Response

Reviewer 1, comment 1

In the introduction (p.2, l.64), the authors describe the various deleterious actions resulting from MR activation in the heart. However, they do not mention the rapid (within days) remodeling of ion channels that occurs in cardiomyocytes upon stimulation with aldosterone. This remodeling leads to a positive chronotropic response of the myocyte beating frequency in vitro and to ventricular arrhythmias in vivo, which are responsible for hyperaldosteronism-related sudden death. This electrical remodeling, involving the (re)expression of low-threshold (T-type) voltage-operated calcium channels, probably contributes to the higher risk of early hospitalization for cardiac fibrillation (and death) that is prevented by MRA therapy (p.14, l.483). Although the beneficial effect was observed mostly with spironolactone and eplerenone, this mechanism of action should be also considered for finerenone.

Response:

We thank the reviewer for this important comment, and we agree that the phenomenon of rapid remodeling of ion channels induced by aldosterone observed in cardiomyocytes needs to be mentioned and referenced accordingly. We have therefore updated the introduction to include some information on this topic (please see tracked changes in the updated version of the manuscript on page 3).

Reviewer 1, comment 2

The references are not evenly cited, some are with only starting page number, others with initial and final pages. References 101, 122 and 124 are apparently incomplete.

Minor point: P.19, l.614: SGLT-2i (not SLGT)

Response:

We thank the reviewer for raising these potential issues with the referencing. We’ve checked through the bibliography and the references are cited as per the style used in PubMed for each journal. Some journals do not provide a start and end page for an article e.g., see references 24 (Muñoz-Durango et al.), 48 (Murea et al.) and 51 (Mansour et al.) in the manuscript bibliography (and cited below [1-3]). The exact reason for this is unclear, but in some cases, it may be because the article is ‘published’ online only rather than appearing in a printed issue of the journal.

There are three ‘placeholder’ references currently in the bibliography. We are waiting for journal acceptance following revisions to original articles for two of these references [4,5], the other article is an Editorial in JASN by one of the authors of the current manuscript which is soon to be published [6]. We hope to be able to update these references soon.

We have also corrected the typing error on page 29 of the updated manuscript.

References

  1. Muñoz-Durango, N.; Fuentes, C.A.; Castillo, A.E.; González-Gomez, L.M.; Vecchiola, A.; Fardella, C.E.; Kalergis, A.M. Role of the renin-angiotensin-aldosterone system beyond blood pressure regulation: molecular and cellular mechanisms involved in end-organ damage during arterial hypertension. International Journal of Molecular Sciences 2016, 17, 797.
  2. Murea, M.; Register, T.C.; Divers, J.; Bowden, D.W.; Carr, J.J.; Hightower, C.R.; Xu, J.; Smith, S.C.; Hruska, K.A.; Langefeld, C.D.; et al. Relationships between serum MCP-1 and subclinical kidney disease: African American-Diabetes Heart Study. BMC nephrology 2012, 13, 148.
  3. Mansour, S.G.; Puthumana, J.; Coca, S.G.; Gentry, M.; Parikh, C.R. Biomarkers for the detection of renal fibrosis and prediction of renal outcomes: a systematic review. BMC nephrology 2017, 18, 72.
  4. Pitt, B.; Agarwal, R.; Anker, S.; Ruilope, L.; Rossing, P.; Ahlers, C.; Stat, D.; Brinker, M.; Joseph, A.; Lambelet, M.; et al. [PLACEHOLDER] Association of finerenone use with reduction in treatment-emergent pneumonia and COVID-19 adverse events among patients with type 2 diabetes and chronic kidney disease: A FIDELITY pooled analysis.
  5. Ruilope, L.; Agarwal, R.; Anker, S.; Filippatos, G.; Pitt, B.; Rossing, P.; Sarafidis, P.; Schmieder, R.E.; Joseph, A.; Mentenich, N.; et al. [PLACEHOLDER] Blood pressure and cardiorenal outcomes with finerenone in chronic kidney disease in type 2 diabetes.
  6. Bakris, G. [PLACEHOLDER] Editorial. JASN.

Reviewer 2 Report

In this review article, authors summarized recent researches investigating the MR antagonism by finerenone. This review is very well-written and covers adequate and sufficient current information about this topic. This review also provides the important insight into potential future clinical indications.

I have only one concern. Figure 2 seems to be the same figure as authors’ previously published articles at NEJM (“Cardiovascular Events with Finerenone in Kidney Disease and Type 2 Diabetes” and “Effect of Finerenone on Chronic Kidney Disease Outcomes in Type 2 Diabetes”). I don’t know it is OK or not (copyright etc) to re-use already published graphs on other journal. If it is OK, I believe this review article should be accepted.

Author Response

Reviewer 2, comment 1

I have only one concern. Figure 2 seems to be the same figure as authors’ previously published articles at NEJM (“Cardiovascular Events with Finerenone in Kidney Disease and Type 2 Diabetes” and “Effect of Finerenone on Chronic Kidney Disease Outcomes in Type 2 Diabetes”). I don’t know it is OK or not (copyright etc) to re-use already published graphs on other journal. If it is OK, I believe this review article should be accepted.

Response:

Thank you to the reviewer for raising this concern regarding copyright for Figure 2. The figures have been sufficiently modified (e.g., the vertical scale on the charts have been altered and the colors used in the figures have been updated), such that copyright permission does not need to be obtained from the original journals.

Reviewer 3 Report

This paper is well written and very interesting. It is a review which well explains the pathophysiology of MR overactivation and its possible counteracting by finerenone, in different organs. The role of genes and biomarkers modulated by finerenone in different clinical models is also reported.

The role of finerenone and its ability to counteract the effects of MR overactivation is very interesting from a clinical point of view, especially regarding the long-term action of finerenone and the reduction of MR-mediated inflammation, ROS generation, endothelial dysfunction,  ECM remodelling, cellular hypertrophy and fibrosis.

There is one organ in particular where the results of long-term inflammation and resulting tissue damage are clinically significantly relevant, with a high prevalence in the aged male population, which is the prostate. Benign prostatic hyperplasia (BPH) is one of the most common phenomena related to the aging process. Recently, the role of chronic inflammation in BPH has been investigated, looking in particular at hyperproliferation, ROS damage and tissue remodelling. A number of new anti-inflammatory drugs, or combination of drugs, have been very recently tested and proven able to reduce prostatic chronic inflammation in vitro. NF-kB actication, ROS production and  IL-6 and IL-8 production have been described in BPH (well reported in Saponaro M., et al, IJMS, 2020; DOI: 10.3390/ijms21239178, a reference to add to the reference list). 

I think the Authors should expand in the Introduction, and also Conclusions section, mentioning  the possible future applications of inhibition of MR overactivation also in other unexplored  systems and organs with high prevalence of chronic inflammatory  disease, such as the prostate. This would make the manuscript very up-to-date and would be and extrimely interesting research proposal and clinical speculation, especially  for scientists and physicians involved with the aging males, like general practioners,  urologists, hospitalists, and geriatricians. 

Author Response

Reviewer 3, comment 1

There is one organ in particular where the results of long-term inflammation and resulting tissue damage are clinically significantly relevant, with a high prevalence in the aged male population, which is the prostate. Benign prostatic hyperplasia (BPH) is one of the most common phenomena related to the aging process. Recently, the role of chronic inflammation in BPH has been investigated, looking in particular at hyperproliferation, ROS damage and tissue remodelling. A number of new anti-inflammatory drugs, or combination of drugs, have been very recently tested and proven able to reduce prostatic chronic inflammation in vitro. NF-kB activation, ROS production and IL-6 and IL-8 production have been described in BPH (well reported in Saponaro M., et al., IJMS, 2020; DOI: 10.3390/ijms21239178, a reference to add to the reference list).

I think the Authors should expand in the Introduction, and also Conclusions section, mentioning the possible future applications of inhibition of MR overactivation also in other unexplored systems and organs with high prevalence of chronic inflammatory disease, such as the prostate. This would make the manuscript very up-to-date and would be and extremely interesting research proposal and clinical speculation, especially for scientists and physicians involved with the aging males, like general practitioners, urologists, hospitalists, and geriatricians.

Response:

We would like to thank the reviewer for this important comment on chronic inflammatory processes associated with BPH and the recommendation to cite Saponaro et al. [7].

We agree that BPH shares several mechanistic features described in our manuscript including inflammation, ROS and tissue remodeling, although a causal role of MR overactivation in BPH has not been identified. Castro et al. conducted an early controlled clinical trial with spironolactone in BPH, which failed to demonstrate more than a marginal benefit from treatment in only two out of 34 variables assessed (symptom score and residual urine)[8]. Experimental clinical use of spironolactone e.g., in acne vulgaris, hirsutism and even prostate cancer (i.e., beyond its original labeled indications for the treatment of edematous conditions) was based on its known antiandrogenic effects as a nonspecific steroid hormone receptor antagonist which potently blocks the androgen receptor (AR) in addition to the MR. Therefore, a clear differentiation between potential AR- and MR-mediated pathophysiological effects was a scientific challenge until a) a more selective MRA (eplerenone) became available and b) the expression profiles of all steroid hormone receptors in all tissues and/or cell types were determined.

In 1997, Hirasawa et al. examined immunolocalization of MR and 11bHSD2 (which confers specificity on MR and aldosterone by inactivating glucocorticoids) in the same cells or tissues of various human exocrine or secretory glands from non-pathological autopsy specimens. They demonstrated that 11bHSD2 protein colocalizes with MR protein in the majority of sodium-transporting epithelia involved in serous secretion with a few exceptions including lacrimal gland, bile ducts, gall bladder and prostate in which neither 11bHSD2 nor MR could be detected [9]. Therefore, a causal pathophysiological role of MR overactivation in human prostate seems unlikely.

Nevertheless, since this reviewer points to an extremely interesting coexisting pathophysiology in BPH based on mechanisms such as inflammation, ROS generation, and hypertrophy, we have updated the introduction to section 3. Beyond the Cardiovascular and Renal Systems: Consequences of MR Overactivation in Other Organs’ to include some information about the potential role of MR overactivation in other organ systems including the prostate (please see tracked changes in the updated version of the manuscript on page 16).

References

7. Saponaro, M.; Giacomini, I.; Morandin, G.; Cocetta, V.; Ragazzi, E.; Orso, G.; Carnevali, I.; Berretta, M.; Mancini, M.; Pagano, F.; et al. Serenoa repens and urtica dioica fixed combination: In-vitro validation of a therapy for benign prostatic hyperplasia (BPH). International Journal of Molecular Sciences 2020, 21, 9178.

8. Castro, J.E.; Griffiths, H.J.; Edwards, D.E. A double-blind, controlled, clinical trial of spironolactone for benign prostatic hypertrophy. Br J Surg 1971, 58, 485-489.

9. Hirasawa, G.; Sasano, H.; Takahashi, K.; Fukushima, K.; Suzuki, T.; Hiwatashi, N.; Toyota, T.; Krozowski, Z.S.; Nagura, H. Colocalization of 11 beta-hydroxysteroid dehydrogenase type II and mineralocorticoid receptor in human epithelia. The Journal of clinical endocrinology and metabolism 1997, 82, 3859–3863.